## [Transparent Peer Review file · Communications Biology]

Structural analysis reveals that water molecules mediate self-activation of GPR99

Corresponding Author: Professor Heng Liu

Version 0:

Reviewer comments:

Reviewer #1

(Remarks to the Author)

The manuscript "Self-activation of GPR99 mediated by water molecule" describes a 2.9Å structure of an important GPCR family member. The authors have designed and performed the experiment to support their structure and hypothesis. Building of water molecules in a 2.9Å cryo-EM map can be sometimes speculative, especially in the case of the membrane proteins, where there is a lot of noise due to detergent molecules. The authors have tried to support their claim with biochemical assays.

1. Rephrase: Line 73, "Despite experimental approaches and molecular dynamics (MD) simulations were used to provide insights into the role of water in GPCR activation²², the atom-level structure is needed to elucidate the crucial role of water molecules as structural determinants."
2. References: line 246, Reference 5 is missing.
3. The referencing is inconsistent in the text. While the reference numbers are superscript on the letter in the main text, in the materials and methods section, they are separated by a space (lines 400, 401, 402 and so on).
4. Inconsistency in the use of °C in the text.
5. The contour levels are not mentioned in any figure, especially when showing water density. Fig. 1C, 3C, 4. Figure S2H should also mention the contour levels.
6. Figure S3: It is common to see densities in the membrane protein structures that look similar to lipids. Did the authors try to build any lipid molecules in them? This can be informative to the readers.
7. A local resolution map showing the built ECL2 loop residues and the water molecules will be more informative. A figure could be added in the supplementary to depict the same.
8. B-factors of water molecules can be mentioned.
9. A multiple sequence alignment showing conservation of ECL2 and the residues interacting with the mentioned water molecules should be added.
10. Table S1 "Magnification 10,500X", check if this is correct.

Reviewer #2

(Remarks to the Author)

The manuscript "Self-activation of GPR99 mediated by water molecules" by Xiao, M et al. reports the Cryo-EM structure of GPR99 (OXGR1) in complex with a MiniGq G-Protein complex. The authors further report that the receptor is activated by ECL2 and activation is also mediated by a network of potentially conserved waters despite having only a 2.9 angstrom resolution map. This being said, I requested the em map and pdb coordinates but only received the em map and a validation file of the submitted pdb, so I am unable evaluate most of the scientific claims within the manuscript. Overall the manuscript is "okay" written and I would have recommended the manuscript to be accepted with some revision considering the competition (see below) but absence of the pdb file precludes this recommendation. Furthermore, it is also noted that there is potentially at least two other groups which also have the cryo-em structure of GPR99 including the released coordinates of PDB ID 8YYW and 8YYX along with an additional four unreleased cryo-em structures including an "apo" structure and thus the mansucfpt may have been hastily written.

However, I've made some recommendations to improve the manuscript:

- 1) the authors claim no em density for the alpha helical domain of the miniGq, understandable because the miniGq and other miniG proteins are engineered to not have the alpha helical domain so this comment should be removed.
- 2) When discussing ECL2, it would be great to have a sequence alignment with GPR52. Additionally, authors provide a figure showing that GPR52b extends further into the orthosteric site but relative depth should also be stated.
- 3) When describing interactions between receptor residues, the distance should be shown in the figures along within the manuscript or at least in a supplemental table. The type of interaction should also be described, ie. backbone to side chain, etc.
- 4) Missing from Figure 3A is the charge potential range and unit.
- 5) The discussion on inactive state is too speculative and can be removed.
- 6) What is the significance of describing the Gprotein interactions, if residues are significant then perhaps mutagenesis along with functional assays would need to be included. Also, are the interactions conserved with similar GPCRs.
- 7) For the identified water network, perhaps subjecting the water locations and network with other tools such as <https://www.nature.com/articles/s41467-025-61315-x>.
- 8) Check references, one of the references says "invalid".

Reviewer #3

(Remarks to the Author)

This study by Xiao et al. presents a high-resolution cryo-EM structure of human GPR99 receptor in complex with miniGq in a ligand-free, constitutively active state. The authors provide structural and functional evidence that the second extracellular loop (ECL2) occupies the orthosteric binding pocket and mediates basal receptor activation. They also identify an extended polar network involving structural water molecules that connects ECL2 to the receptor core, offering mechanistic insight into GPR99 self-activation. Although the manuscript provides important observations about the ECL2 mediated self-activation in GPR99, I have following concerns:

1. Structural comparison of GPR99's ECL2 with other constitutively active GPCRs is valuable, particularly the identification of a short helix occupying the orthosteric pocket and its stabilization by water molecules. However, the functional implications of ECL2's shallower projection in GPR99 remains elusive and needs to be clarified. The authors should clarify how this structural difference impacts G protein coupling or activation kinetics.
2. The authors report that disruption of the second ECL2 segment reduces basal activity (Figure 2C, D); however, it remains unclear whether these effects stem from the loss of helical structure, disrupted water coordination, or weakened interactions with the receptor core. Clarifying this distinction would strengthen the mechanistic interpretation.
3. Authors present an interesting and novel hypothesis that structural water molecules, in conjunction with ECL2, cooperatively stabilize the active conformation of GPR99 and promote its basal signaling. The concept of water-mediated receptor activation is intriguing and could provide new insight into mechanisms of constitutive activity in GPCRs. However, no information is given on the resolution at which these waters were confidently placed, or on the criteria used to assign them.
4. The claim that water overlaps well with succinate in GPR91 (Fig. 3B) is intriguing but appears speculative. From the figure, the water molecules seem to reside near, rather than directly overlapping, the succinate. The authors should be more cautious in this interpretation and clarify the spatial relationship to avoid overstatement.
5. Is the rearrangement of ECL2 sufficient for full receptor activation, or is it stabilizing a pre-activated state? TM6 outward and TM7 inward shifts are consistent with canonical GPCR activation. However, the claim that "TM5 shows a significant replacement" is vague.
6. To strengthen the mechanistic understanding of GPR99 self-activation, the authors should include a sequence alignment of ECL2 with other constitutively active GPCRs (e.g., GPR52, GPR21, BILF1) to identify conserved motifs such as agonist-like motifs (ALMs). Additionally, elaborating on the conformational transition of ECL2—from a β -sheet in the inactive model to a helical or unstructured form in the active state—would clarify its structural role in activation. A brief discussion on the physiological relevance of GPR99's constitutive activity under native conditions would also enhance the broader significance of the findings.

Minor comments:

1. In the Results section (lines 88–89), the statement that GPR99 possesses intrinsic self-activation similar to GPR17, GPR21, GPR52, and BILF1 would be strengthened by including experimental controls using one of these receptors or, at a minimum, citing relevant studies that directly demonstrate their constitutive activity. Additionally, in lines 106–109, when describing how ECL2 projects into the orthosteric pocket, it would be helpful to clarify that these receptors belong to class A GPCRs, where ECL2-mediated self-activation has been recurrently observed as a class-specific mechanism. This context would help situate GPR99 within a broader structural and functional framework.
2. The structural comparison in Figure 2A would benefit from greater clarity. It is currently difficult to interpret due to overlapping models and an unclear color scheme. To improve readability, I recommend displaying the individual GPCR structures separately before showing their superimposition with GPR99. A distinct color coding for each receptor, and particularly for the water molecules (which should be clearly labelled as originating from the GPR99 structure), would help orient readers. Additionally, please clarify in the figure legend whether the water molecules are modelled or experimentally observed, and which structure they are associated with. Furthermore, the rationale for selecting GPR52 for structural comparison should be explicitly stated. Ideally, structural overlays of all four mentioned GPCRs (GPR17, GPR21, GPR52, and BILF1) with GPR99 should be shown to support claims of structural similarity and ECL2-mediated activation. Lastly, the text describes the middle segment of ECL2 adopting a short helix that acts as an agonist-like motif (ALM) and a second segment functioning as a lid over the orthosteric pocket. This important feature should be clearly illustrated with a zoomed-in

view showing the ECL2 conformational elements and the binding pocket in the same frame for better visualization and mechanistic interpretation.

3. In line 113–114, the manuscript notes that ECL2 occupies the orthosteric pocket of GPR99 with a shallower projection compared to GPR52, attributed to differences in residue composition. However, this key structural distinction would benefit from clearer visual representation. Please consider enhancing the figure panel(s) by highlighting the specific residues responsible for this difference and their spatial arrangement. A side-by-side comparison or zoomed-in view of ECL2 in both GPR99 and GPR52, with annotated residues, would be especially helpful.

4. Regarding the agonist-like motif (ALM), it remains unclear whether a similar motif is present in GPR99. The authors should elaborate on this in the main text and ideally support it with a sequence alignment of the ECL2 region across relevant GPCRs (e.g., GPR17, GPR52, GPR21, BILF1), including GPR99, to evaluate conservation of ALM features.

5. In Figure 3B, to convincingly demonstrate the alignment of water molecules with succinate, additional orientations of the overlay would be valuable. Presenting views from different angles would better support the proposed structural mimicry and enhance clarity for the reader.

Version 1:

Reviewer comments:

Reviewer #1

(Remarks to the Author)

The authors have responded to all my comments and have revised their manuscript, which is now improved.

Minor comment:

1. Authors have built Cholesterol and CHS into the densities. It will be useful to add a couple of relevant references to show that they may stabilize the membrane protein structures.

Reviewer #2

(Remarks to the Author)

I would like to thank the authors for addressing all of the other reviewers' comments. I realize this was a significant amount of additional work but feel this has greatly improved the manuscript. This being said, I would like to recommend the current revised manuscript for acceptance in the journal without further edits on my behalf.

Reviewer #3

(Remarks to the Author)

In this revised version the authors have responded to all my previous comments. The revisions have improved the clarity, and I endorse the manuscript for publication.

Point-by-point response (COMMSBIO-25-4995)

We would like to thank all the reviewers for their thorough evaluation of our manuscript and for their constructive suggestions, which helped us to improve the quality of our work. We addressed all reviewers' concerns by substantially revising the manuscript and adding new figures and discussions. Please find a detailed response to each reviewer's comments below, while all the changes in the manuscript are highlighted in yellow.

Reviewer #1 (Remarks to the Author):

The manuscript "Self-activation of GPR99 mediated by water molecule" describes a 2.9Å structure of an important GPCR family member. The authors have designed and performed the experiment to support their structure and hypothesis. Building of water molecules in a 2.9Å cryo-EM map can be sometimes speculative, especially in the case of the membrane proteins, where there is a lot of noise due to detergent molecules. The authors have tried to support their claim with biochemical assays.

We thank the Reviewer for the time to review our manuscript.

1. Rephrase: Line 73, "Despite experimental approaches and molecular dynamics (MD) simulations were used to provide insights into the role of water in GPCR activation²², the atom-level structure is needed to elucidate the crucial role of water molecules as structural determinants."

Following the suggestion of the Reviewer, we have rephrased the sentence:

"Although experimental approaches and molecular-dynamics simulations have shed light on the role of water in GPCR activation, atomic-level structures are still required to establish water molecules as definitive structural determinants."

2. References: line 246, Reference 5 is missing.

We thank the Reviewer for spotting this point. We have corrected it.

3. The referencing is inconsistent in the text. While the reference numbers are superscript on the letter in the main text, in the materials and methods section, they are separated by a space (lines 400, 401, 402 and so on).

We thank the Reviewer for spotting this point. We have corrected them.

4. Inconsistency in the use of °C in the text.

Thank you for spotting this. We have checked the manuscript to correct inconsistency.

5. The contour levels are not mentioned in any figure, especially when showing water density. Fig. 1C, 3C, 4. Figure S2H should also mention the contour levels.

We thank the reviewer for reminding us of this point. We have added the contour levels along with density presentation.

6. Figure S3: It is common to see densities in the membrane protein structures that look similar to lipids. Did the authors try to build any lipid molecules in them? This can be informative to the readers.

Following the Reviewer's suggestion, we build Cholesterol and Cholesteryl hemisuccinate molecules into the densities, which usually enhance the stability of GPCRs. The contour levels are presented as well in new Fig. S5.

7. A local resolution map showing the built ECL2 loop residues and the water molecules will be more informative. A figure could be added in the supplementary to depict the same.

Following the Reviewer's suggestion, we have added the densities of ECL2, pocket residues and water molecules in Fig. S3. We also compared the densities in different contour levels, which showed that the density of waters is comparable to receptor's density. The comparison indicates that isolated densities in orthosteric pocket are waters rather than noises.

8. B-factors of water molecules can be mentioned.

We thank the reviewer for reminding us of this point. We have included B-factors in Table S1.

9. A multiple sequence alignment showing conservation of ECL2 and the residues interacting with the mentioned water molecules should be added.

We thank the Reviewer for the suggestion. We have added sequence alignment and structure comparison in Fig. S4. We found that GPR99 exhibits a distinct sequence and structure feature from GPR52 and GPR21. GPR99 and GPR91 are closely related with conservation of ECL2. The residues interacting with waters are highlighted with blue stars in Fig. S4c. The constitutive activity assay of GPR91 in Fig. 1a indicates its self-activation mediated by water molecules as GPR99.

Furthermore, we also summarize two different conformations of ECL2 in self-activated GPCRs (Fig. S4e). In conformation 1, at the bottom of ECL2, the C-terminal part of ECL2 participates in the interactions with waters. However, in conformation 2, the N-terminal part of ECL2 inserts deeply into the orthosteric binding pocket to activate the receptor as ALM.

10. Table S1 "Magnification 10,500X", check if this is correct.

Thank you for catching this typo. It should be 105,000X. We have corrected it.

Reviewer #2 (Remarks to the Author):

The manuscript "Self-activation of GPR99 mediated by water molecules" by Xiao, M et al. reports the Cryo-EM structure of GPR99 (OXGR1) in complex with a MiniGq G-Protein complex. The authors further report that the receptor is activated by ECL2 and activation is also mediated by a network of potentially conserved waters despite having only a 2.9 angstrom resolution map. This being said, I requested the em map and pdb coordinates but only received the em map and a validation file of the submitted pdb, so I am unable evaluate most of the scientific claims within the manuscript. Overall the manuscript is "okay" written and I would have recommended the manuscript to be accepted with some revision considering the competition (see below) but absence of the pdb file precludes this recommendation. Furthermore, it is also noted that there is potentially at least two other groups which also have the cryo-em structure of GPR99 including the released coordinates of PDB ID 8YYW and 8YYX along with an additional four unreleased cryo-em structures including an "apo" structure and thus the mansucfipt may have been hastily written.

We thank the Reviewer for taking the time to provide useful and warranted feedback on our manuscript. We have made corrections as discussed below and uploaded the map and model with this revision.

However, I've made some recommendations to improve the manuscript:

1) the authors claim no em density for the alpha helical domain of the miniGq, understandable because the miniGq and other miniG proteins are engineered to not have the alpha helical domain so this comment should be removed.

We thank the Reviewer for spotting this point. We have corrected it.

2) When discussing ECL2, it would be great to have a sequence alignment with GPR52. Additionally, authors provide a figure showing that GPR52b extends further into the orthosteric site but relative depth should also be stated.

Following the Reviewer's suggestion, we have added sequence alignment and structure comparison with GPR91, GPR17, BILF1, GPR52 and GPR21 in Fig. 2a and Fig. S4. We give the extending depth (the distance to mircoswitch 6.48) of ECL2s of GPR52 in updated Fig. 2b. We found that GPR99 exhibits a distinct sequence and structure feature from GPR52 and GPR21. GPR99 and GPR91 are closely related with conservation of ECL2. The residues interacting with waters are highlighted in Fig. S4c. The constitutive activity assay of GPR91 in Fig. 1a indicates its self-activation mediated by water molecules as GPR99.

Furthermore, we also summarize two different conformations of ECL2 in self-activated GPCRs (Fig. S4e). In conformation 1, at the bottom of ECL2, the C-terminal part of ECL2 participates in the interactions with waters. However, in conformation 2, the N-terminal part of ECL2 inserts deeply into the orthosteric binding pocket to activate the receptor as ALM.

3) When describing interactions between receptor residues, the distance should be shown in the figures along within the manuscript or at least in a supplemental table. The type of interaction should

also be described, ie. backbone to side chain, etc.

Following the Reviewer's suggestion, we have reorganized the Fig. 3c and provided the hydrogen bonds distance.

4) Missing from Figure 3A is the charge potential range and unit.

We have added the charge potential range and unit.

5) The discussion on inactive state is too speculative and can be removed.

We agree with the Reviewer that AlphaFold-predicted structures cannot necessarily represent the real protein conformations. We have removed some speculative discussion on inactive state. To better understand the active mechanism of GPR99 mediated by water molecules, we performed a structure comparison of our structure with oxoglutarate-bound active GPR99 (PDB: 8YYW) and the inactive structure of rat GPR91 (6RNK) in Fig. 4 and 5. The comparison of the key motifs and structure elements, which play crucial roles in GPCR activation, shows that our structure adopts a classic active conformation. Therefore, we add new comparison of crucial structure elements between our GPR99 and GPR91 (6RNK) in Fig. 4 and 5.

Following the Reviewer's suggestion, we tried to discuss the active mechanism appropriately to avoid overstatement.

6) What is the significance of describing the Gprotein interactions, if residues are significant then perhaps mutagenesis along with functional assays would need to be included. Also, are the interactions conserved with similar GPCRs.

GPR99 and GPR91 are closely related with high sequence similarity. They can both couple G_q protein and exhibit a certain similarity in the coupling mode. The residue R3.50 in the DRY motif establishes hydrogen bonding interactions with the $\alpha 5$ -helix backbone of G_q protein, playing a key role in G protein coupling. Here, we confirmed the polar interactions between GPR99 and G_q through nanoBiT-based assays. Additionally, the hydrophobic interactions between side chains of I235, L236, L240, and L245 in the $\alpha 5$ -helix of G_q and the hydrophobic residues in GPR99 is also found in GPR91 (ref: <https://doi.org/10.1038/s41422-024-00968-7>).

7) For the identified water network, perhaps subjecting the water locations and network with other tools such as <https://www.nature.com/articles/s41467-025-61315-x>.

We thank the Reviewer for the suggestion. First, we used Phenix douse to identified and modeled water molecules into the globular densities in the orthosteric pocket. The water molecules were then confirmed by the Metric Ion Classification (MIC) tool as suggested by the Reviewer. The model was then checked for accuracy and refined in COOT and Phenix against the sharpened map from cryoSPARC. We updated the Methods section and cited the reference (<https://www.nature.com/articles/s41467-025-61315-x>).

8) Check references, one of the references says "invalid".

We thank the Reviewer for spotting this point. We have corrected it.

Reviewer #3 (Remarks to the Author):

This study by Xiao et al. presents a high-resolution cryo-EM structure of human GPR99 receptor in complex with miniGq in a ligand-free, constitutively active state. The authors provide structural and functional evidence that the second extracellular loop (ECL2) occupies the orthosteric binding pocket and mediates basal receptor activation. They also identify an extended polar network involving structural water molecules that connects ECL2 to the receptor core, offering mechanistic insight into GPR99 self-activation. Although the manuscript provides important observations about the ECL2 mediated self-activation in GPR99, I have following concerns:

We thank the reviewer for taking the time to provide useful feedback and detailed evaluation on our manuscript.

1. Structural comparison of GPR99's ECL2 with other constitutively active GPCRs is valuable, particularly the identification of a short helix occupying the orthosteric pocket and its stabilization by water molecules. However, the functional implications of ECL2's shallower projection in GPR99 remains elusive and needs to be clarified. The authors should clarify how this structural difference impacts G protein coupling or activation kinetics.

We thank the Reviewer for spotting this point. We have performed sequence alignment and structure comparison of GPR99 with GPR52, GPR21, GPR17, BILF1 and GPR91 in Fig. 2a and Fig. S4.

GPR99 and GPR91 are closely related with conservation of ECL2. The residues interacting with waters are highlighted in Fig. S4c. The constitutive activity assay of GPR91 in Fig. 1a indicates its self-activation mediated by water molecules as GPR99. However, we found that GPR99 exhibits a distinct sequence and structure feature from GPR52 and GPR21. ECL2 of GPR52 and GPR21 extend deeply further into the orthosteric site with a close distance to microswitch residue 6.48 (Fig. 2b and Fig. S4a), which allow ECL2 can act as ALM. In GPR99 and GPR91, the distance from ECL2 and residue 6.48 is over 15 Å, which makes it impossible to activate the receptor through direct contact with residue 6.48. Therefore, the coordination of waters in GPR99 extends the depth of ECL2 into orthosteric pocket to activate receptor, which mimics the succinate in GPR91 (PDB: 8YKW). Although we don't have an apo GPR91-G protein structure, we speculate that the high basal activity of GPR91 is mediated by waters as that found in GPR99 (Fig. 1a).

2. The authors report that disruption of the second ECL2 segment reduces basal activity (Figure 2C, D); however, it remains unclear whether these effects stem from the loss of helical structure, disrupted water coordination, or weakened interactions with the receptor core. Clarifying this distinction would strengthen the mechanistic interpretation.

We thank the Reviewer for the suggestion. In our structure, the second ECL2 segment is involved in direct and indirect interaction with water molecules, and interaction with TM2/3/6/7 and ECL1 of GPR99 (updated Fig. 3c). The alanine substitution of 185-188^{ECL2} and deletion of 186-187^{ECL2} resulted in integrated effects on the basal activity of GPR99. Following the Reviewer's suggestions, we have clarified this in manuscript.

3. Authors present an interesting and novel hypothesis that structural water molecules, in conjunction with ECL2, cooperatively stabilize the active conformation of GPR99 and promote its basal signaling. The concept of water-mediated receptor activation is intriguing and could provide new insight into mechanisms of constitutive activity in GPCRs. However, no information is given on the resolution at which these waters were confidently placed, or on the criteria used to assign them.

Following the Reviewer's suggestion, we have added the densities of ECL2 and water molecules in Fig. S3. We also compared the densities in different contour levels, which showed that the density of waters is comparable to receptor's density. The comparison means that isolated densities in orthosteric pocket are waters rather than noises.

4. The claim that water overlaps well with succinate in GPR91 (Fig. 3B) is intriguing but appears speculative. From the figure, the water molecules seem to reside near, rather than directly overlapping, the succinate. The authors should be more cautious in this interpretation and clarify the spatial relationship to avoid overstatement.

We thank the Reviewer for the detailed inspection. We agree that the water molecules don't directly overlap with the succinate. We have corrected the description in our manuscript and present the local structure in two different angles (Fig. 3b).

5. Is the rearrangement of ECL2 sufficient for full receptor activation, or is it stabilizing a pre-activated state? TM6 outward and TM7 inward shifts are consistent with canonical GPCR activation. However, the claim that "TM5 shows a significant replacement" is vague.

In our structure, ECL2 occupied the orthosteric pocket and formed extensive interactions with receptor directly or mediated by water molecules (Fig. 3c). As a result, the extracellular part of receptor underwent conformational changes, and propagated to cytoplasmic end. In the structural comparison with inactive GPR91 (6RNK) structure and GPR91-G_q structure (PDB: 8YKW), the conserved motifs DR^{3.50}Y, C^{6.47}F^{6.48}X^{P6.50}, N^{7.49}L^{7.50}LLY^{7.53} and P^{5.50}I^{3.40}F^{6.44} exhibited active conformation. Especially, TM6 outward and TM7 inward shifts, the hallmarks of the classical activation hallmarks in class A GPCRs, were also observed (Fig. 4b, c). Given that GPR99 can bind G_q protein and exhibits a high level of constitutive activity without an agonist (Fig. 1a), it is likely that the ECL2 is functionally equivalent to an agonist to fully activate GPR99 in the presence of waters. Moreover, we also compare our structure with an oxoglutarate-bound active GPR99, which was released in PDB when our manuscript was in preparation. Our structure shows a full active conformation as oxoglutarate-bound active

GPR99 with RMSD 0.34 Å (Fig. 4a). Overall, the rearrangement of ECL2 coordinated with water molecules is sufficient to fully activate GPR99.

We also modified the description of TM5 replacement on activation:

“Moreover, TM5 also shows a significant outward replacement at cytoplasmic end upon activation (Fig. 4b, c).”

6. To strengthen the mechanistic understanding of GPR99 self-activation, the authors should include a sequence alignment of ECL2 with other constitutively active GPCRs (e.g., GPR52, GPR21, BILF1) to identify conserved motifs such as agonist-like motifs (ALMs). Additionally, elaborating on the conformational transition of ECL2—from a β -sheet in the inactive model to a helical or unstructured form in the active state—would clarify its structural role in activation. A brief discussion on the physiological relevance of GPR99’s constitutive activity under native conditions would also enhance the broader significance of the findings.

Following the Reviewer’s suggestion, we have performed sequence alignment and structure comparison of ECL2 with GPR91, GPR52, GPR21, GPR17 and BILF1 in Fig. 2a and Fig. S4. We didn’t find a conserved agonist-like motifs (ALMs) in GPR99 as described in GPR52. Instead, we found the residues of ECL2 involved in GPR99 activation are conserved in GPR91 (Fig. S4c), which is also confirmed by the constitutive activity of GPR91 (Fig. 1a).

We agree that elaborating on the conformational transition of ECL2 would strengthen the functional role in receptor activation. However, note that the AlphaFold- predicted inactive GPCR structures may not be entirely precise. Therefore, our proposed conformational changes in ECL2 from a β -sheet to a helical or unstructured form only represent a plausible hypothesis. We would like not discuss the conformational changes of ECL2 to avoid overstatement.

To better understand the active mechanism of GPR99 mediated by water molecules, we performed a structure comparison of our structure with oxoglutarate-bound active GPR99 (PDB: 8YYW) and the inactive structure of rat GPR91 (6RNK) in Fig. 4 and 5. The comparison of the key motifs and structure elements, which play crucial roles in GPCR activation, shows that our structure adopts a classic active conformation. Therefore, we add new comparison of crucial structure elements between our GPR99 and GPR91 (6RNK) in Fig. 4 and 5.

Furthermore, we also summarize two different conformations of ECL2 in self-activated GPCRs (Fig. S4e). In conformation 1, at the bottom of ECL2, the C-terminal part of ECL2 participates in the interactions with waters. However, in conformation 2, the N-terminal part of ECL2 inserts deeply into the orthosteric binding pocket to activate the receptor as ALM.

We agree that GPR99’s constitutive activity plays a physiological role in native conditions. Following the Reviewer’s suggestion, we added a discussion in the manuscript in Discussion section:

“Highly expressed in nasal mucosal vascular smooth muscle cells and in nasal polyposis with robust constitutive activity, GPR99 is likely to maintain the airway epithelium in a standby state: it preserves baseline goblet-cell abundance, primes the mucosecretory machinery, and enables

low concentration LTE4 to trigger massive mucus discharge. Moreover, by sensing local fluctuations of the TCA-cycle intermediate 2-oxoglutarate, GPR99 couples metabolic status to low-threshold inflammatory amplification, thereby providing a pre-configured platform that rapidly escalates immune output upon infection or allergen encounter. However, more further studies are needed to elucidate the physiological relevance of GPR99's constitutive activity under native conditions.”

Minor comments:

1. In the Results section (lines 88–89), the statement that GPR99 possesses intrinsic self-activation similar to GPR17, GPR21, GPR52, and BILF1 would be strengthened by including experimental controls using one of these receptors or, at a minimum, citing relevant studies that directly demonstrate their constitutive activity. Additionally, in lines 106–109, when describing how ECL2 projects into the orthosteric pocket, it would be helpful to clarify that these receptors belong to class A GPCRs, where ECL2-mediated self-activation has been recurrently observed as a class-specific mechanism. This context would help situate GPR99 within a broader structural and functional framework.

Following the Reviewer’s suggestion, we have added GPR17’s constitutive activity assay as control (Fig. 1a). And we also add references (refs 12-16) to demonstrate the constitutive activity of GPR21, GPR52, and BILF1.

We thank the Reviewer’s suggestion. As described above, we have added a description of two different conformations of ECL2 in self-activated GPCRs (Fig. S4e).

2. The structural comparison in Figure 2A would benefit from greater clarity. It is currently difficult to interpret due to overlapping models and an unclear color scheme. To improve readability, I recommend displaying the individual GPCR structures separately before showing their superimposition with GPR99. A distinct color coding for each receptor, and particularly for the water molecules (which should be clearly labelled as originating from the GPR99 structure), would help orient readers. Additionally, please clarify in the figure legend whether the water molecules are modelled or experimentally observed, and which structure they are associated with. Furthermore, the rationale for selecting GPR52 for structural comparison should be explicitly stated. Ideally, structural overlays of all four mentioned GPCRs (GPR17, GPR21, GPR52, and BILF1) with GPR99 should be shown to support claims of structural similarity and ECL2-mediated activation. Lastly, the text describes the middle segment of ECL2 adopting a short helix that acts as an agonist-like motif (ALM) and a second segment functioning as a lid over the orthosteric pocket. This important feature should be clearly illustrated with a zoomed-in view showing the ECL2 conformational elements and the binding pocket in the same frame for better visualization and mechanistic interpretation.

We thank the Reviewer’s suggestion. We have updated Fig. 2 to present a zoomed-in view of ECL2 and waters.

Following the Reviewer’s suggestion, we have added the densities of ECL2, pocket residues and water molecules in Fig. S3. We also compared the densities in different contour levels,

which showed that the density of waters is comparable to receptor's density. The comparison means that isolated densities in orthosteric pocket are waters rather than noises.

We have performed sequence alignment and structure comparison of ECL2 with GPR91, GPR52, GPR21, GPR17 and BILF1 in Fig. 2a and Fig. S4. Moreover, the structure comparison (Fig. S4e) shows that the conformation of ECL2 in these receptors can be assigned to two distinct families based on the sequence and structure.

3. In line 113–114, the manuscript notes that ECL2 occupies the orthosteric pocket of GPR99 with a shallower projection compared to GPR52, attributed to differences in residue composition. However, this key structural distinction would benefit from clearer visual representation. Please consider enhancing the figure panel(s) by highlighting the specific residues responsible for this difference and their spatial arrangement. A side-by-side comparison or zoomed-in view of ECL2 in both GPR99 and GPR52, with annotated residues, would be especially helpful.

We thank the Reviewer's suggestion. We have given a sequence comparison of ECL2 in GPR99 and GPR52 in new Fig. S4. We also updated the Fig. 2 to present a zoomed-in view of ECL2. The sequence and structure comparison show that the distinct spatial structure stems from the globe different residues composition rather than specific residues.

4. Regarding the agonist-like motif (ALM), it remains unclear whether a similar motif is present in GPR99. The authors should elaborate on this in the main text and ideally support it with a sequence alignment of the ECL2 region across relevant GPCRs (e.g., GPR17, GPR52, GPR21, BILF1), including GPR99, to evaluate conservation of ALM features.

Following the Reviewer's suggestion, we have performed a sequence alignment of ECL2 with GPR52, GPR21, GPR17 and BILF1. We didn't find a conserved agonist-like motifs (ALMs) as described in these receptors. Instead, we found the residues of ECL2 involved in GPR99 activation are conserved in GPR91 (Fig. S4), which is also confirmed by the constitutive activity of GPR91.

We also summarize two different conformations of ECL2 in self-activated GPCRs (Fig. S4e). In conformation 1, at the bottom of ECL2, the C-terminal part of ECL2 participates in the interactions with waters. However, in conformation 2, the N-terminal part of ECL2 inserts deeply into the orthosteric binding pocket to activate the receptor as ALM.

5. In Figure 3B, to convincingly demonstrate the alignment of water molecules with succinate, additional orientations of the overlay would be valuable. Presenting views from different angles would better support the proposed structural mimicry and enhance clarity for the reader.

We thank the Reviewer's suggestion. We have updated Fig. 3b, and present a zoomed-in view of water molecules and succinate from different angles.

Point-by-point response (COMMSBIO-25-4995)

We would like to thank Reviewer #1 for the suggestion to improve the quality of our work. We addressed the reviewer's concern and added new references. Please find a detailed response to the reviewer's comment below, while all the changes in the manuscript are highlighted in yellow.

Reviewer #1 (Remarks to the Author):

Minor comment:

1. Authors have built Cholesterol and CHS into the densities. It will be useful to add a couple of relevant references to show that they may stabilize the membrane protein structures.

We thank the Reviewer for the suggestion. We have added three references (refs 30-32) in the third paragraph of Discussion section.